# Glycine Supplementation in Obesity Worsens Glucose Intolerance through Enhanced Liver Gluconeogenesis

**DOI:** 10.3390/nu15010096

**Published:** 2022-12-24

**Authors:** Anaïs Alves, Frédéric Lamarche, Rémy Lefebvre, Eva Drevet Mulard, Arthur Bassot, Stéphanie Chanon, Emmanuelle Loizon, Claudie Pinteur, Aline Maria Nunes de Lira Gomes Bloise, Murielle Godet, Gilles J. P. Rautureau, Baptiste Panthu, Béatrice Morio

**Affiliations:** 1CarMeN laboratory, UMR INSERM U1060/INRAE U1397, Université Claude Bernard Lyon 1, Université de Lyon, 69310 Pierre-Bénite, France; 2Laboratory of Fundamental and Applied Bioenergetics, INSERM U1055, Université Grenoble Alpes, 38400 Saint Martin d’Hères, France; 3ICBMS CNRS U5246, Université Claude Bernard Lyon 1, Université de Lyon, 69622 Villeurbanne, France; 4Erika Cosset Team, Cancer Research Centre of Lyon, UMR INSERM U1052/CNRS 5286, 69008 Lyon, France; 5Department of Physical Education and Sport Sciences, Laboratory of Nutrition, Physical Activity and Phenotypic Plasticity, Universidade Federal de Pernambuco, UFPE, 55604-000 Vitória de Santo Antão, PE, Brazil; 6Centre de Résonance Magnétique Nucléaire à Très Hauts Champs, UMR CNRS U5082/ENS Lyon, Université Claude Bernard Lyon 1, Université de Lyon, 69100 Villeurbanne, France

**Keywords:** dietary supplement, amino acid metabolism, obesity-related metabolic disorders, insulin resistance, mitochondria

## Abstract

Interactions between mitochondria and the endoplasmic reticulum, known as MAMs, are altered in the liver in obesity, which contributes to disruption of the insulin signaling pathway. In addition, the plasma level of glycine is decreased in obesity, and the decrease is strongly correlated with the severity of insulin resistance. Certain nutrients have been shown to regulate MAMs; therefore, we tested whether glycine supplementation could reduce insulin resistance in the liver by promoting MAM integrity. Glycine (5 mM) supported MAM integrity and insulin response in primary rat hepatocytes cultured under control and lipotoxic (palmitate 500 µM) conditions for 18 h. In contrast, in C57 BL/6 JOlaHsd mice (male, 6 weeks old) fed a high-fat, high-sucrose diet (HFHS) for 16 weeks, glycine supplementation (300 mg/kg) in drinking water during the last 6 weeks (HFHS-Gly) did not reverse the deleterious impact of HFHS-feeding on liver MAM integrity. In addition, glycine supplementation worsened fasting glycemia and glycemic response to intraperitoneal pyruvate injection compared to HFHS. The adverse impact of glycine supplementation on hepatic gluconeogenesis was further supported by the higher oxaloacetate/acetyl-CoA ratio in the liver in HFHS-Gly compared to HFHS. Although glycine improves MAM integrity and insulin signaling in the hepatocyte in vitro, no beneficial effect was found on the overall metabolic profile of HFHS-Gly-fed mice.

## 1. Introduction

Hepatic insulin resistance is a major metabolic disorder that causes a heavy burden on the healthcare system as it affects 60–80% of obese and type 2 diabetic patients [1]. Interestingly, obese and type 2 diabetic patients show abnormal circulating amino acid profiles [2] that are particularly characterized by decreased plasma glycine concentration [3,4]. The decrease in circulating glycine concentration is associated with the severity of insulin resistance [5,6]. Several prospective studies showed that a decrease in plasma glycine concentration is a strong predictive factor for incident glucose intolerance and type 2 diabetes [7,8,9]. Glycine is a non-essential amino acid with many metabolic roles. However, the involvement of glycine in metabolic diseases associated with obesity is still poorly understood, and part of the mechanisms responsible for its metabolic effects have not yet been elucidated [10].

Hepatic insulin resistance is associated with metabolic inflexibility, mitochondrial dysfunction, and endoplasmic reticulum (ER) stress, which suggests a potential relationship between hepatic metabolic disorders and impaired functions of both organelles. Mitochondria and ER communicate through contact points called mitochondria associated-ER membranes (MAMs) which dynamically regulate cellular functions [11]. MAM integrity has been associated with metabolic homeostasis in the liver [12]. Specifically, MAMs have been identified as crucial platforms involved in the regulation of the hepatic insulin response [13,14,15,16]. Furthermore, MAM integrity is disrupted and MAM flexibility is lost in the liver of obese and type 2 diabetic mice [13]. Therefore, MAMs are now recognized as crucial hubs of cell signaling, and targeting MAMs in the liver could be a novel strategy to prevent hepatic insulin resistance.

Based on the alteration of MAMs and the variation of glycine concentration in the serum of obese and diabetic patients, we hypothesized that glycine may be involved in the regulation of hepatic MAM integrity, and we anticipated that restoring glycine availability in obese mice may restore hepatic insulin resistance by supporting MAM integrity in the liver. Indeed, glycine interacts with one-carbon metabolism through the enzyme glycine N-methyltransferase (GNMT), thereby potentially regulating methylation processes which can regulate gene and protein expression [10,17]. GNMT catalyzes the methylation of glycine using S-adenosylmethionine to form sarcosine with the concomitant production of S-adenosylhomocysteine. Therefore, it plays an important role in the balance of methyl groups in the liver [10,17].

In this study, we explored in vitro, on primary rat hepatocytes, and in vivo, in diet-induced obese mice, whether glycine supplementation could positively impact on the hepatic insulin response by promoting MAM integrity, thus contributing to improved glucose homeostasis.

## 2. Material and Methods

### 2.1. Primary Rat Hepatocytes

Primary hepatocytes were isolated using a modified collagenase perfusion method according to Berry et al. [18] and Groen et al. [19]. Primary rat hepatocytes were plated on collagen type I-coated culture plates and cultured in Dulbecco’s Modified Eagle’s Medium (DMEM, PAA Laboratories, Velizy-Villacoublay, France) with 3 g/L glucose at 37 °C and in a humid atmosphere with 5% CO_2_. DMEM was supplemented with 2 mM glutamine, 2 mM antibiotic/antimycotic, 1% penicillin streptomycin solution, 7.5 mM DL-sodium lactate, and 10% fetal calf serum (FCS), which is the standard medium. The percentage of viability was determined using the trypan blue exclusion method (on an average 85–90% of the viable cells). After 4 h of adhesion, hepatocytes were cultured for 18 h in control (standard medium + bovine serum albumin, BSA) or lipotoxic (standard medium + palmitate 500 μM) supplemented or not with glycine (5 mM).

### 2.2. Animal Study

This animal study was performed in accordance with the French guidelines for the care and use of animals [20] and approved by the regional ethic committee (CECCAP LS_2022_001). Thirty 5-week-old male C57 BL/6 JOlaHsd mice (Envigo, France) were housed in groups of five on sawdust bedding in plastic cages in an enriched environment up to the age 5 months. Artificial lighting was provided on a fixed 12 h light–dark cycle with ad libitum access to standard chow and water. Mice were fed a standard chow diet (SD, n = 10, 35.6% carbohydrates, 10.2% fat by mass, Genobios, France) for 16 weeks or a high-fat, high-sucrose diet (HFHS, n = 20, 36% fat and 17% sucrose by mass, Envigo, France) for 10 weeks [21]. Then, the HFHS-fed mice were divided into 2 groups of 10 animals and remained on the HFHS diet for an additional 6 weeks. The first group continued to receive water, while the second group received glycine in water (300 mg/kg/d). An oral glucose tolerance test (OGTT, 1.5 g/kg body weight) was performed on 4 h-fasted mice one week before the end of the protocol. Five days apart, a pyruvate tolerance test (1.5 g/kg body weight, injected intraperitoneally, ipPTT) was performed on overnight-fasted mice. In both tests, blood glucose was monitored with a glucometer for 2 h before and at 15, 30, 45, 60, 90, and 120 min after gavage or injection. Glucose and pyruvate tolerance were evaluated based on the blood glucose area under the curve (AUC) [21]. Finally, overnight-fasted mice were euthanized with elongation, and the liver and leg muscles were quickly removed, weighted, and divided for further analyses. 

### 2.3. Exploration of the Insulin Signaling Pathway

In vitro, primary hepatocytes were depleted in serum for 3 h and then incubated in DMEM without FBS in the absence or presence of insulin (10-7 M) for 15 min. In vivo, overnight-fasted mice were injected with either NaCl or insulin (0.75 U/kg body weight) 15 min before the sacrifice. The liver and *gastrocnemius* muscles were rapidly removed and frozen in liquid nitrogen. Insulin-stimulated Ser473 phosphorylation (#4060, Cell Signaling) of protein kinase B (Akt, #4691, Cell Signaling) was assessed using western blotting in primary rat hepatocytes, mouse liver, and *gastrocnemius* muscle.

Additional information was obtained from a preliminary study performed following the same protocol (regional ethic committee approval, CECCAP LS_2017_004) in which peripheral insulin response was evaluated in 4 h-fasted animals. Four hour-fasted mice (n = 10 per group) were injected intraperitoneally with insulin (0.75 U/kg body weight). Blood glucose was monitored with a glucometer before and at 15, 30, and 45 min after injection. Insulin tolerance was evaluated based on the blood glucose area over the curve (AOC) [21].

### 2.4. In Situ Proximity Ligation Assay (PLA)

ER-mitochondrial interactions were assessed using in situ PLA which targeted the complex between the ER inositol 1,4,5-triphosphate receptor (IP3 R)-1 (ab5804, Abcam, Paris, France) and the mitochondrial voltage-dependent anion channel (VDAC)-1 (ab14734, Abcam, UK) as previously described [13]. In vitro, the assays were conducted on 4% paraformaldehyde-fixed and 0.1% triton-permeabilized primary rat hepatocytes using a green, fluorescent in situ PLA DUOLINK kit (Merck, Darmstadt, Germany). In situ, the assays were carried on in 4 µm paraffin sections of 4% paraformaldehyde-fixed and paraffin-embedded mouse liver samples using a bright-field in situ PLA DUOLINK kit (Merck, USA). Images were analyzed using a custom written Fiji macro on 10 images/sample in 3 to 5 independent series, and the number of VDAC1-IP3 R1 dots was expressed per nucleus.

### 2.5. Triglyceride and Glycogen Assay

Mouse liver samples were embedded in an optimal cutting temperature (OCT) compound and frozen in liquid nitrogen. Oil Red O (ORO-k-250, Biognost) staining was performed in 10 μm cryosections to quantify the liver triglycerides, and 4 µm paraffin sections were stained with PAS diastase (PAD-2-IFU, Clinisciences) to measure glycogen content. Images were analyzed using a custom-written Fiji macro, and data expressed the percentage of the area. Furthermore, liver triglyceride, oxaloacetate, and acetyl-CoA concentrations were assayed using the commercial assay kits, including Triglycerides GPO Method (Biolabo, Maizy, France), Oxaloacetate Colorimetric/Fluorometric, and PicoProbe™Acetyl-CoA Fluorometric (CliniSciences, Nanterre, France), respectively. Results were expressed relative to the protein concentration, and the oxaloacetate to acetyl-CoA ratio was calculated.

### 2.6. Western Blot

Proteins of primary hepatocytes or mouse livers were lysed in a RIPA buffer (150 mM NaCl, 1.0% IGEPAL^®^ CA-630, 0.5% sodium deoxycholate, 0.1% SDS, 50 mM Tris, pH 8.0). Protein concentration of the samples was quantified using the Bradford protein assay (BioRad, Marnes-la-Coquette, France). Protein content was quantified after SDS-PAGE electrophoresis (10 or 15% acrylamide gels) migration using the following antibodies: anti-Akt (4691 L, Cell signaling, USA), anti-S473-Phospho AKT (4060 L, Cell Signaling, Saint-Cyr-L’École, France), and anti-tubulin (T5168, Merck, USA). The signal was quantified with ImageLab software (Biorad, USA). Data were expressed relative to tubulin before calculating the phospho-Akt/Akt ratio and the insulin-induced fold-change in Akt phosphorylation.

### 2.7. Gene Expression

Total RNA was extracted using a Trizol Reagent kit following the method of Chomczynski and Sacchi [22] as previously decribed [23]. RNA concentration was measured using the Nanodrop 2000 (Thermo Fisher Scientific, Waltham, MA USA) according to the manufacturer’s recommendations. Reverse transcription (RT) of RNA to complementary DNA (cDNA) was performed using 1 μg of total RNA in 10 μL of total reaction volume of PrimeScript reagents RT kit. Total RNA samples were treated with DNase to prevent the potential contamination with genomic DNA. Gene expression of *gnmt* was explored by qPCR (quantitative real time Polymerase Chain Reaction) using SYBR^®^ Green mix and Rotor-Gene 6200 (Corbett Research, Gentaur, Paris, France) and normalized using TBP (TATA-binding protein) mRNA. Primers sequences for *gnmt* were 5′ GAAGGAGCCAGCCTTTGACA 3′ and 3′ AGGTGAGCAAAACTGTTCCC 5′ for rats and 5′ GGAAAGAGCCATCCTTTGAC 3′ and 3′ GCAAGTGAGCAAAACTGTTCC 5′ for mice. Primers sequence for TBP were 5′TGGTGTGCACAGGAGCCAAG 3′ and 3′ TTCACATCACAGCTCCCCAC 5′ for rats and 5′ TGGTGTGCACAGGAGCCAAG 3′ and 3′ TTCACATCACAGCTCCCCAC 5′ for mice.

### 2.8. Mice Liver and Muscle Endometabolome Analyses through ^1^ H-NMR Spectroscopy

Metabolites were extracted using 100% methanol and a Precellys Homogeneizer as previously described [24]. NMR spectra were obtained as previously described [24] on a Bruker Avance III spectrometer operating at a ^1^ H frequency of 800.14 MHz and equipped with a 5 mm TXI probe. Identification of the metabolites was carried out based on the ^1^ H 1 D NMR data with ChenomX NMR Suite 8.0 software (ChenomX Inc., Edmonton, Canada) and confirmed based on analyses of 2 D ^1^ H–^1^ H TOCSY, ^1^ H–^13^ C HSQC, and J-Resolved NMR spectra. Metabolite concentrations were determined based on the ^1^ H 1 D NOESY experiments using ChenomX. A pure lactate solution (1 g/L, Fisher) was used as a concentration reference and exploited using the ERETIC2 utility from TopSpin to add a digitally synthesized peak to a spectrum [25].

### 2.9. Statistical Analysis

Statistical analysis was performed using GraphPad Prism^®^ software v8 (GraphPad Software, San Diego, CA, USA). All data were presented as means ± standard error of the mean (SEM). The normality of the data was assessed using Shapiro–Wilk or Agostino and Pearson omnibus normality tests, and homogeneity was assessed using Bartlett’s test. Two-way ANOVA followed by a Student post-hoc test were used to explore the effect of glycine in control and lipotoxic conditions for the in vitro experiments. The three groups were compared using one-way ANOVA followed by a Sidak’s post-hoc test or Kruskal–Wallis test for the in vivo experiments. The level of significance was set at *p* < 0.05.

## 3. Results

### 3.1. Glycine Enhances MAM Contact Sites and Insulin Response in Primary Rat Hepatocytes 

In primary rat hepatocytes, palmitate (500 µM) for 18 h induced a 38.5% higher *gnmt* mRNA content compared to BSA (*p* < 0.01), whereas glycine (5 mM) had no significant impact in both the control and lipotoxic conditions (Figure 1A). Palmitate caused a 26.5% lower number of VDAC1-IP3 R1 contact points per cell compared to BSA. In contrast, glycine induced a 32.5 and 50.8% higher number of VDAC1-IP3 R1 contact points per cell compared to untreated cells in both the control and lipotoxic conditions, respectively (*p* < 0.001, Figure 1B,C). As expected [13,16], these positive impacts were associated with an improved insulin response, especially under control conditions (*p* < 0.05, Figure 1D,E). We thus tested whether glycine supplementation could improve liver MAM integrity and insulin response in diet-induced obese mice and contribute to improved glucose homeostasis.

### 3.2. Characteristics of Diet-Induced Obese Mice 

The HFHS diet induced 43.8% and 57.1% heavier body and liver weights compared to SD at the end of the study, respectively (*p* < 0.01). These differences were not altered after glycine supplementation, so HFHS-Gly animals evidenced similar body and liver weights as HFHS (Table 1). The liver triglyceride content was 81.4% higher and glycogen content 32.4% lower in HFHS compared to SD (*p* < 0.05, Figure 2A–E). Glycine supplementation did not significantly alter these parameters (Figure 2A–E).

The liver metabolic fingerprint was explored using ^1^ H-NMR metabolomics. The HFHS diet was associated with a 21.7% lower glycine concentration compared to SD (*p* < 0.01), and HFHS-Gly tended to restore the value to the SD level (*p* = 0.068 vs. HFHS, Figure 2F,G). HFHS-Gly was associated with an 86.0% higher sarcosine concentration compared to SD (*p* < 0.05, Figure 2G and Appendix A). Related to the use of glycine for sarcosine synthesis, the liver mRNA content in *gnmt* mRNA was 31.2% lower in HFHS compared to SD (*p* < 0.01) and was not significantly altered by glycine supplementation (Figure 2H). Additionally, choline and o-phosphocholine were 2 to 3-fold higher in both HFHS and HFHS-Gly compared to SD (*p* < 0.01, Figure 2G). Finally, the HFHS diet was also associated with a 22.1% to 26.1% lower concentration in N-acetylcysteine and glycerol compared to SD, respectively (*p* < 0.05, Figure 2G). These alterations were no longer significant following glycine supplementation compared to SD values (Figure 2G). 

### 3.3. Glycine Supplementation Does Not Influence Liver MAM Integrity and Insulin Response in Diet-Induced Obese Mice

As already described [13], liver MAM integrity assessed using in situ PLA was disrupted in HFHS compared to SD. Indeed, the number of VDAC1-IP3 R1 contact points per nucleus was 33.3% lower in HFHS compared to SD (*p* < 0.05, Figure 3A,B). The difference did not reach the level of significance for the VDAC1-IP3 R2 interaction (Figure 3C,D). Glycine supplementation did not modify the HFHS-induced alterations (Figure 3B–D). Exploration of insulin-induced Akt phosphorylation showed that HFHS was associated with a 40.6% lower fold-change in Akt phosphorylation compared to SD (*p* < 0.05), while HFHS-Gly tended to restore values to the SD level (Figure 3E,F).

### 3.4. Glycine Supplementation Worsens Glucose Homeostasis of Diet-Induced Obese Mice

We further investigated in vivo glucose homeostasis. Four-hour fasting glycemia was 23.2% and 25.3% higher in HFHS and HFHS-Gly compared to SD, respectively (*p* < 0.0001, Figure 4C). In contrast, while the differences between HFHS and SD values did not reach the level of significance regarding overnight-fasting glycemia, overnight-fasting glycemia of HFHS-Gly was 41.4% higher than in SD (*p* < 0.001, Figure 4E). In vivo tests evidenced a deterioration in glucose and pyruvate tolerance in HFHS-fed animals compared to SD (Figure 4A–F). AUC_OGTT_ and AUC_ipPTT_ were 110.2% (*p* < 0.001, Figure 4A,C) and 53.6% (*p* < 0.05, Figure 4D,F) higher in HFHS compared to SD. In HFHS-Gly-fed animals, the glycemia response to OGTT was similar to HFHS (Figure 4A); therefore, glycine supplementation did not significantly alter AUC_OGTT_ compared to HFHS (Figure 4C). In contrast, glycemia response to _ip_PTT was 24.4% higher than that of HFHS 15 min after pyruvate injection (*p* < 0.01) and remained higher than that of HFHS for the rest of the test (*p* = NS, Figure 4D). However, overall, the impact of glycine supplementation on AUC_ipPTT_ did not reach the level of significance compared to HFHS (Figure 4F). AUC_ipPTT_ of HFHS-Gly-fed animals was 87.5% higher compared to SD (*p* < 0.05, Figure 4F). We assessed the concentration of oxaloacetate that drives gluconeogenesis to better understand the discrepancy between ipPTT and insulin responses in the livers of HFHS-Gly-fed animals. The oxaloacetate to acetyl-CoA ratio was 96.7% higher in HFHS compared to SD (*p* < 0.0001, Figure 4G) and a further 43.0% higher in HFHS-Gly compared to HFHS (*p* < 0.001, Figure 4G). This finding shows that the glycine supplementation enhanced gluconeogenesis activity in the liver of HFHS-Gly-fed animals compared to HFHS.

### 3.5. The Impact of Glycine Supplementation on the Skeletal Muscle of Diet-Induced Obese Mice

We further phenotyped the *gastrocnemius* muscle status since skeletal muscles are key tissues involved in glycine [26] and glucose [27] homeostasis. Glycine concentration in the muscle was similar between HFHS and SD and 34.6% higher in HFHS-Gly compared to the other two groups (*p* < 0.001, Figure 5A). Interestingly, concentrations in sarcosine and glutamate were 25.0% and 25.7% lower and alanine was 19.2% higher in HFHS compared to SD, respectively (*p* < 0.01, Figure 5B and Appendix A). In HFHS-Gly, leucine and glycerol concentrations were 20.3% and 38.6% higher compared to SD, respectively (*p* < 0.05, Figure 5B and Appendix A). Finally, alanine and lactate concentrations were 23.3% and 30.2% lower in HFHS-Gly compared to HFHS, respectively (*p* < 0.001, Figure 5B and Appendix A).

Exploration of insulin-induced Akt phosphorylation evidenced that HFHS and HFHS-Gly both tended to decrease the insulin-induced Akt phosphorylation compared to SD, but the difference did not reach the level of significance (Figure 5C,D). These results were corroborated by the insulin tolerance test performed in a separate but similar protocol (Figure 5E–G). Four-hour fasting glycemia was 17.5% and 25.0% higher in HFHS and HFHS-Gly compared to SD, respectively (*p* < 0.05, Figure 5E). However, while both HFHS and HFHS-Gly tended to decrease AOC_ipITT_ compared to SD, neither reached the level of significance (Figure 5F,G). These results showed that skeletal muscle responded to glycine supplementation, but these adaptations did not alter the insulin resistance.

## 4. Discussion

Our previous study on diet-induced obese mice showed that the lowest plasma glycine concentration in obesity is also observed in the liver [24]. In that organ, this study showed that the lower availability of glycine was partially compensated by glycine supplementation. Although glycine positively impacted MAM integrity and insulin response in vitro, glycine supplementation had little or no beneficial impact on these parameters in the liver of diet-induced obese mice. It also had no apparent beneficial impact on skeletal muscle insulin resistance associated with diet-induced obesity. Unexpectedly, glycine supplementation promoted hepatic gluconeogenesis, thereby contributing to the worsening of overnight-fasting glycemia.

Our work is the first to evaluate the impact of glycine on liver MAM integrity. In situ PLA has proven its reliability for exploring VDAC1-IP3 R contact points in vitro and in vivo [12,13,15,16]. The discrepancy between the in vitro and in vivo results suggests that glycine implication may be context dependent and involve one or more of the multiple metabolic [10]. In particular, as glycine metabolism has been shown to be important for skeletal muscle in obesity [26], the discrepancy between the in vitro and in vivo results suggests that crosstalk between skeletal muscle and the liver may play a key role in determining the availability and metabolic fate of glycine in the hepatocytes. The divergence might also involve the expression of GNMT which is involved in the regulation of S-adenosyl-methionine levels and methylation processes. *Gnmt* gene expression was indeed enhanced by palmitate in vitro and reduced in obesity in vivo. In addition, the mechanisms may be more complex than the regulation of S-adenosyl-methionine levels as several compensatory mechanisms can be activated according to the *gnmt* gene expression level and the availability of metabolites interacting with the one-carbon cycle [28]. Given the correlation between low plasma glycine concentration and the severity of insulin resistance in obesity and type 2 diabetes [10], we hypothesized that glycine might have bioactive functions. Few studies have evidenced that liver MAM integrity can be regulated by nutrients. The deciphered signaling pathways involve either glucose and the protein phosphatase 2 A (PP2 A) [12] or arginine, the synthesis of nitric oxide (NO), and the protein kinase G (PKG) [16]. Palmitate has also been identified to alter MAM integrity in vitro in hepatocytes [13,14] and myotubes [29], but the exact mechanisms are still unknown. Further studies are needed to follow up on these findings. However, the data collected in our study do not encourage further work on glycine.

The present work is also the first that evaluated the impact of glycine on liver insulin response, and it found no beneficial impact on nutritional dosage. It is the second work that observed that glycine supplementation at a dose below 0.3 g/kg/d for 6 to 10 weeks does not improve muscle insulin resistance and glucose tolerance in an animal model of obesity [26]. No recent clinical interventions have been published using dietary glycine in patients with obesity and/or cardiometabolic disorders. A former clinical work in elderly patients with HIV evidenced that 0.1 g of glycine/kg/day for 14 days in association with N-acetylcysteine improved insulin sensitivity [30], potentially due to improved glutathione synthesis and antioxidant protection. Our data showed that glutathione content was not altered following glycine supplementation in the liver and *gastrocnemius* muscle, which contrasts with this clinical trial, potentially because our dosage was about four times lower (i.e., a human equivalent dose of 24.3 mg/kg/day, [31]) and not associated with N-acetylcysteine. In addition, our study did not show a beneficial effect of glycine supplementation on liver biochemistry, especially on triglyceride content, which contrasts with Zhou et al. [32]. The authors used a 23-fold higher dose of glycine (i.e., 3.5 g/kg/d, the human equivalent dose being 565 mg/kg/day, [31]) for 24 weeks in diet-induced obese rats [32]. In this study, glycine supplementation decreased triglyceride levels and protected against HFHS-induced non-alcoholic steatohepatitis, but the authors did not explore its impact on glucose homeostasis. Therefore, it would be interesting to study the dose-response to glycine supplementation in obesity starting from nutritional doses to therapeutic doses. To complete the exploration, it would also be relevant to explore the impact of the form of administration of glycine, i.e., administered in water or food as opposed to gavage. Based on current knowledge, our work showed that in the liver, like glycerol, lactate, and other amino acids, such as alanine, the carbon skeleton of glycine supplied at a nutritional dosage can be metabolized to pyruvate to support gluconeogenesis through the synthesis of phosphoenolpyruvate [33]. The present study provides evidence that gluconeogenesis is strongly upregulated in obesity, which may imply that supplementary dietary glycine is integrated in gluconeogenesis and routed toward glucose synthesis. Interestingly, overnight fasting glycemia and the early glycemic peak in response to the pyruvate tolerance test were the most impacted following glycine supplementation compared to other measurements (i.e., glucose and insulin tolerance tests and insulin response in tissues). This suggests that providing glycine in drinking water chronically correlates with hepatic gluconeogenesis induction but, as observed through Akt-phosphorylation assay, it does not worsen tissue insulin resistance.

NMR exploration interestingly evidenced differences between liver and muscle metabolic fingerprints in response to dietary manipulations. In the liver, HFHS was associated with a lower glycine concentration which was restored close to SD values due to glycine supplementation in parallel to enhanced sarcosine content. This may indicate that glycine supplementation enhances its conversion to sarcosine through GNMT. In contrast, in *gastrocnemius* muscle, HFHS did not alter glycine concentration, but it was associated with lower content in sarcosine and glutamate and higher content in leucine and alanine compared to SD. These observations corroborate a recent demonstration that muscle glycine metabolism in obesity is shifted to provide carbon for the pyruvate-alanine cycle in a manner regulated by branched chain amino acids [26]. Furthermore, HFHS may divert glycine metabolism from GNMT activity. Interestingly, glycine supplementation induced a strong increase in glycine content in *gastrocnemius* muscle, which contrasts with a partial restoration of sarcosine and glutamate concentration, and it induced a significant decrease in alanine and lactate content. This raises two issues that need to be further assessed. Firstly, a question arises whether glycine supplementation is sufficient to restore GNMT activity in skeletal muscle. Secondly, it would be important to explore whether glycine supplementation in obesity alters the Cori and Cahill cycles between skeletal muscle and the liver, thereby contributing to increased hepatic gluconeogenesis.

In conclusion, the present study did not find a coherent impact of glycine on hepatocyte and liver MAM integrity, which supports the idea that glycine does not consistently interact with pathways involved in the regulation of MAM integrity in the liver. Alternative or compensatory mechanisms may contribute to loosening the link between glycine and the signaling pathways involved in regulating MAM integrity. This could explain why glycine can have positive impact on that parameter in vitro and not in vivo. In addition, our observations do not support the beneficial impact of glycine supplementation in obesity to prevent insulin resistance and improve glucose intolerance. This raises the question whether glycine supplementation in obesity could alter alanine and lactate fluxes between skeletal muscles and the liver, which would promote gluconeogenesis and hyperglycemia. Therefore, the long-term effect of glycine supplementation on the risk of developing type 2 diabetes needs to be debated.

## Figures and Tables

**Figure 1 nutrients-15-00096-f001:**
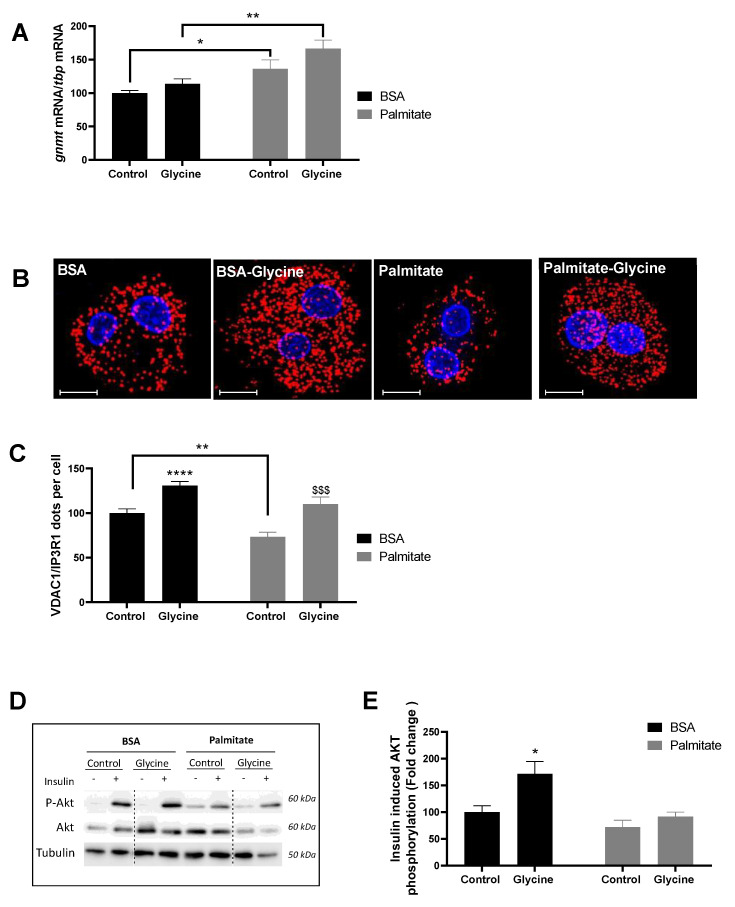
The impact of glycine on MAM integrity and insulin response in primary rat hepatocytes. Primary rat hepatocytes were treated with glycine (5 mM) for 18 h under basal (BSA) or lipotoxic (Palmitate, 500 µM) conditions. (**A**) Glycine N-methyltransferase (Gnmt) mRNA expression relative to TATA box binding protein (Tbp) mRNA, (**B**) representative images (scale = 10 µm) and (**C**) quantitative analysis of VDAC1/IP3R1 interactions per cells using in situ Proximity Ligation Assay (n = 3 series per group, n = 10 images/condition; VDAC1: Voltage Dependent Anion Channel 1; IP3R1: Inositol 1,4,5-trisphosphate receptor type1). (D) representative images and (E) quantitative analysis of insulin (10-7M)-induced Akt phosphorylation (n = 6 series per group). Comparison of all groups vs. BSA: * *p* < 0.05, ** *p* < 0.01, **** *p* < 0.0001; Comparison of palmitate + glycine vs. Palmitate: $$$ *p* < 0.001.

**Figure 2 nutrients-15-00096-f002:**
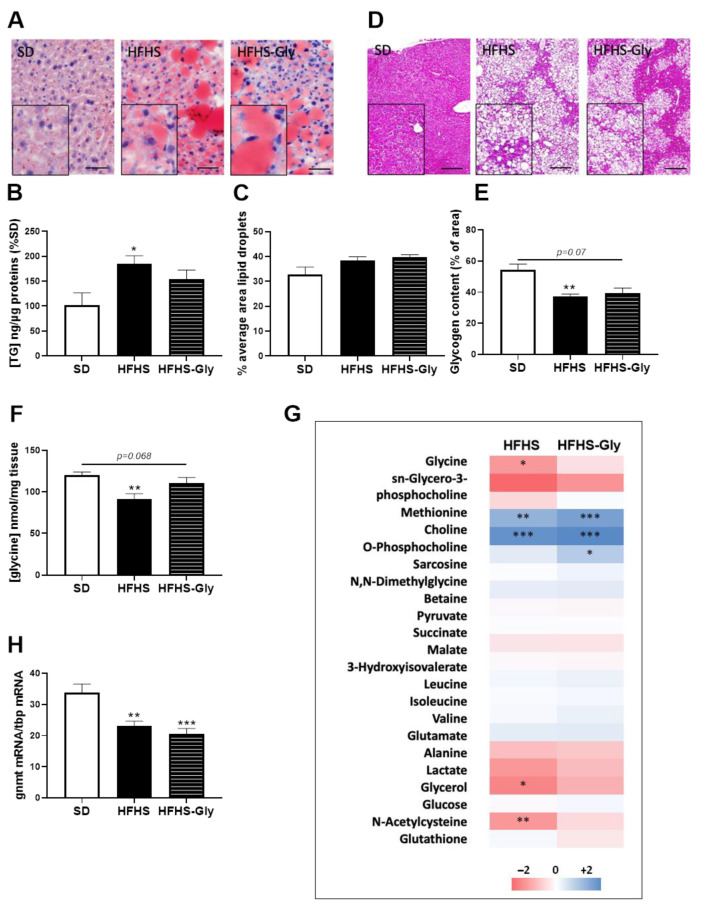
The impact of obesogenic diet and glycine on hepatic metabolism. C57Bl6JOlaHsd mice were fed a standard diet (SD) or a high-fat high-sucrose diet (HFHS) for 16 weeks. Glycine supplementation (300 mg/kg/day) was given in drinking water during the last 6 weeks to half of the HFHS-fed mice (HFHS-Gly). (**A**) representative images and quantitative analysis of liver (scale = 200µm) (**B**) content in triglycerides (TG) and (**C**) mean lipid droplet area (n = 5 mice per group, n = 10 images/mouse). (**D**) representative images (scale = 200µm) and (**E**) quantitative analysis of liver glycogen content (n = 5 mice per group, n = 10 images/mouse). Quantitative NMR analysis of liver (**F**) glycine concent and (**G**) major metabolites (n = 5 mice per group). HFHS and HFHS-Gly data were normalised to the SD group. (**H**) Glycine N-methyltransferase (Gnmt) mRNA expression relative to TATA box binding protein (Tbp) mRNA (n = 10 mice per group). Comparison of HFHS and HFHS-Gly vs. SD: * *p* < 0.05, ** *p* < 0.01, *** *p* < 0.001.

**Figure 3 nutrients-15-00096-f003:**
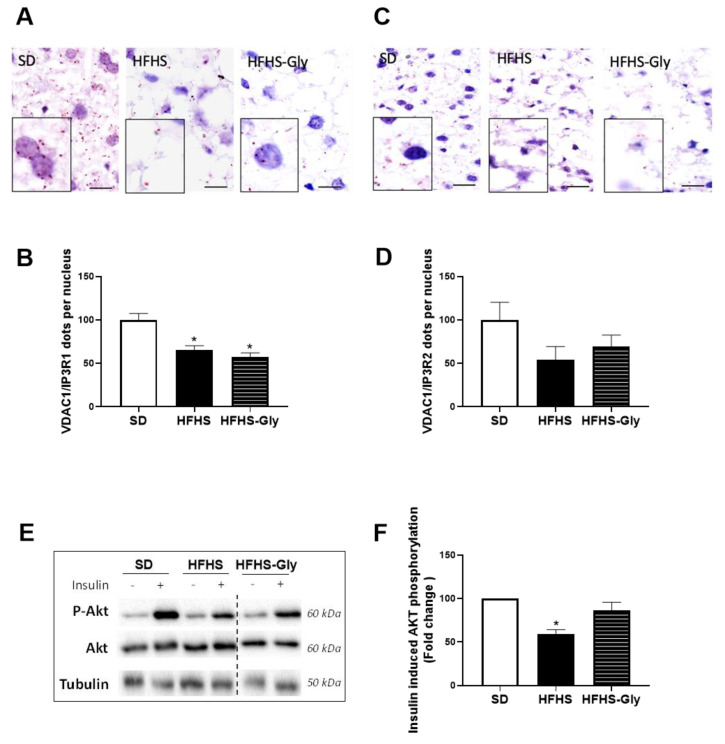
The impact of obesogenic diet and glycine on hepatic MAM integrity and insulin response. C57Bl6JOlaHsd mice were fed a standard diet (SD) or a high-fat high-sucrose diet (HFHS) for 16 weeks. Glycine supplementation (300 mg/kg/day) was given in drinking water during the last 6 weeks to half of the HFHS-fed mice (HFHS-Gly). (**A**) representative images (scale = 10µm) and (**B**) quantitative analysis of VDAC1/IP3R1 contacts per nucleus (n = 5 mice per group, n = 10 images/mouse). (**C**) representative images (scale = 10µm) and (**D**) quantitative analysis of VDAC1/IP3R2 contacts per nucleus (n = 5 mice per group, n = 10 images/mouse). (**E**) Representative images and (**F**) quantitative analysis of insulin (0.75 U/kg)-induced Akt phosphorylation (n = 5 unstimulated and 5 insulin-stimulated mice per group). Comparison of HFHS and HFHS-Gly vs. SD: * *p* < 0.05.

**Figure 4 nutrients-15-00096-f004:**
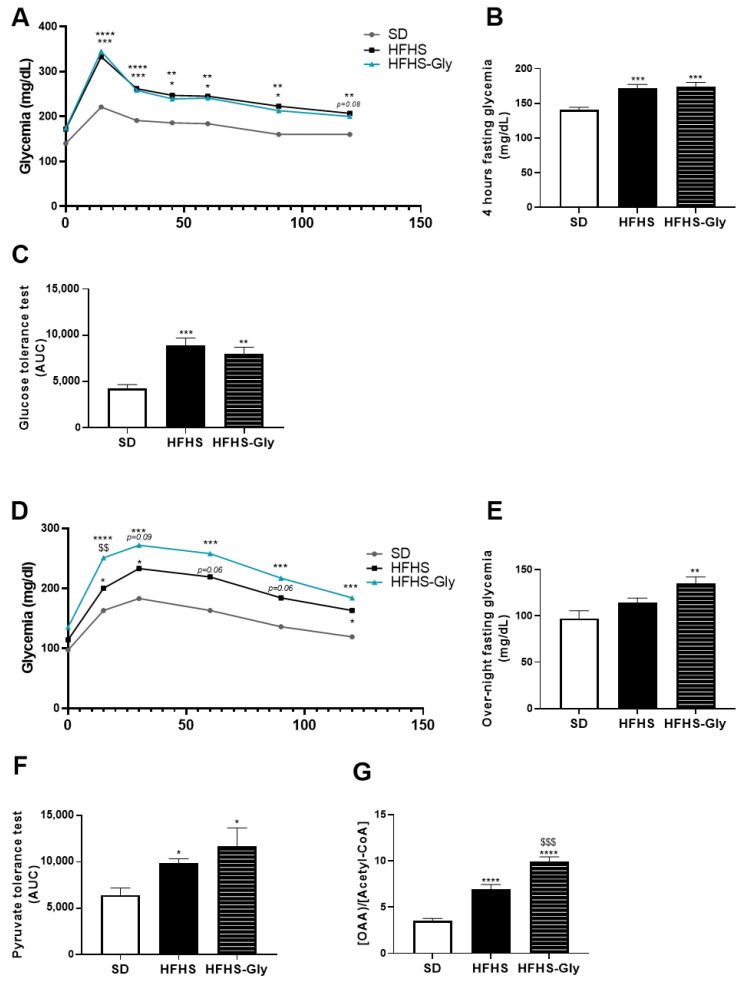
The impact of obesogenic diet and glycine on glucose and pyruvate tolerance tests. C57 Bl6 JOlaHsd mice were fed a standard diet (SD) or a high-fat, high-sugar diet (HFHS) for 16 weeks. Glycine supplementation (300 mg/kg/day) was given in drinking water during the last 6 weeks to half of the HFHS-fed mice (HFHS-Gly). (**A**) shows the glycemic response, while (**B**) highlights 4 h-fasting glycemia, and (**C**) displays the area under the curve (AUC) of the glycemic response to glucose (1.5 g/kg) gavage (OGTT, n = 10 mice per group). (**D**) shows the glycemic response, while (**E**) displays overnight-fasting glycemia, and (**F**) highlights the area under the curve (AUC) of the glycemic response to pyruvate (1.5 g/kg) intraperitoneal injection (ipPTT, n = 10 mice per group). Finally, (**G**) displays the oxaloacetate (OAA) ratio to acetyl-coA in the liver (n = 10 mice per group). Comparison of HFHS and HFHS-Gly vs. SD used * *p* < 0.05, ** *p* < 0.01, *** *p* < 0.001, and **** *p* < 0.0001. Comparison of HFHS-Gly vs. HFHS used $$ *p* < 0.01, $$$ *p* < 0.001.

**Figure 5 nutrients-15-00096-f005:**
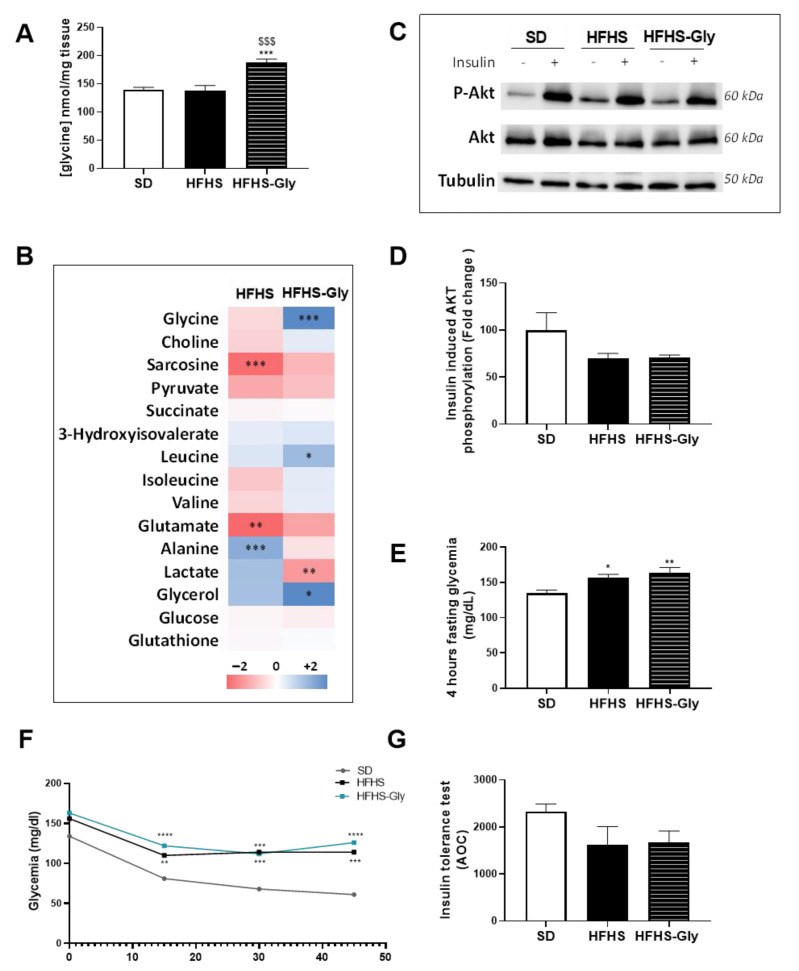
The impact of obesogenic diet and glycine on muscle metabolism and insulin response. C57 Bl6 JOlaHsd mice were fed a standard diet (SD) or a high-fat, high-sugar diet (HFHS) for 16 weeks. Glycine supplementation (300 mg/kg/day) was given in drinking water during the last 6 weeks to half of the HFHS-fed mice (HFHS-Gly). (**A**,**B**) show quantitative NMR analyses of *gastrocnemius* muscle content in glycine and major metabolites (n = 5 mice per group), respectively. HFHS and HFHS-GLY data were normalized to the SD group. (**C**) includes representative images, and (**D**) highlights a quantitative analysis of insulin (0.75 U/kg)-induced Akt phosphorylation (n = 5 unstimulated and 5 insulin-stimulated mice per group). (**E**) displays four-hour fasting glycemia, while (**F**) shows the glycemic response, and (**G**) includes the area over the curve (AOC) of the glycemic response to insulin (0.75 U/kg) intraperitoneal injection (ipITT, n = 10 mice per group). Comparison of HFHS and HFHS-Gly vs. SD used * *p* < 0.05; ** *p* < 0.01; *** *p* < 0.001, and **** *p* < 0.0001. Comparison of HFHS-Gly vs. HFHS used $$$ *p* < 0.001.

**Table 1 nutrients-15-00096-t001:** Animal characteristics at sacrifice.

	SD (n = 10)	HFHS (n = 10)	HFHS-Gly (n = 10)
Body weight (g)	28.8 ± 0.5	41.5 ± 1.3 ***	43.8 ± 1.0 ***
Liver weight (g)	1.19 ± 0.03	1.85 ± 0.19 **	1.91 ± 0.16 **
*Gastrocnemius* muscles (mg)	399 ± 12	399 ± 12	413 ± 9

C57Bl6JOlaHsd male mice were fed a standard diet (SD) or a high-fat high-sucrose diet (HFHS) for 16 weeks. Glycine supplementation (300 mg/kg/day) was given in drinking water during the last 6 weeks to half of the HFHS-fed mice (HFHS-Gly). Comparison of all groups vs. SD used ** *p* < 0.01 and *** *p* < 0.001.

## Data Availability

Contact the corresponding author.

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
