# Peer review of "Glycine Supplementation in Obesity Worsens Glucose Intolerance through Enhanced Liver Gluconeogenesis"

_nutrients, 2022, doi:10.3390/nu15010096_

Round 1

Reviewer 1 Report

Thank you very much for this interesting work on glucose metabolism from a in vitro to in vivo model.

First overall: Please improve the figures by adding more precise abbreviationas and figure explanations. The figure explanations are not really clear to the reader, because the used font size does not clearly differentiate between text and figure captions.

Sometimes it is not really clear which diet group is the main reference group in the analyses. Please be more precise.

The font sizes also changes within the text and the citation style does not seem to be correct.

The in vivo model is a mouse model so a mouse cell line is correct for the in vitro tests, but have the authors also tried to include a test with human liver cells?

The result descrition could be improved by a clearer description of the figure captions in the figures and by an information which is the reference group.

Did the authors think that the different dosages of glycine would impact the results? Or the form of application?

Author Response

We appreciate your positive assessment of the manuscript. Please find bellow our answers to your comments :

First overall: Please improve the figures by adding more precise abbreviation as and figure explanations. The figure explanations are not really clear to the reader, because the used font size does not clearly differentiate between text and figure captions.

The figures have been improved to improve their reading.

Sometimes it is not really clear which diet group is the main reference group in the analyses. Please be more precise.

The clarity of the figure legends has been improved.

The font sizes also changes within the text and the citation style does not seem to be correct.

The font size has been homogenised and the citation style corrected throughout the manuscript.

The in vivo model is a mouse model so a mouse cell line is correct for the in vitro tests, but have the authors also tried to include a test with human liver cells?

The authors sought to work on human liver cells. However, the commercially available cell lines (HuH7, HepG2, HepaRG) are derived from cancer cells, which do not express the key enzyme for glycine metabolism, GNMT. This approach was therefore not possible in this work.

The result description could be improved by a clearer description of the figure captions in the figures and by an information which is the reference group.

The clarity of the figure legends has been improved.

Did the authors think that the different dosages of glycine would impact the results? Or the form of application?

It is indeed possible that glycine dosage may have an impact on the outcome as suggested by our contrasting results with Zhou et al. (2016). These authors used a dosage equivalent to 50 to 60g/d in humans, a dose used in patients with schizophrenia or refractory obsessive-compulsive disorder. We chose to work with a low dose, compatible with a human supplementation for preventing metabolic disorders (# 2g/d) and in agreement with the previous studies in rat (White et al., 2020) and human (Nguyen et al., 2014).  Regarding the form of application, giving glycine in water or in food would lead to similar results as mice eat and drink throughout the day. Different results could have been found if glycine had been given by gavage. We have therefore added this relevant comment to the discussion (page 15).

Reviewer 2 Report

Only minor comments>

The authors should better explain the discranpency between in vivo and in vitro experiments. The authors should suggest a putative or alternative mechanism that might explain those discranpensy. What about the organ cross talk?

Author Response

The authors should better explain the discrepancy between in vivo and in vitro experiments. The authors should suggest a putative or alternative mechanism that might explain those discranpensy. What about the organ cross talk?

Thank you very much for your constructive comments. The discussion has been enhanced accordingly with the proposal : "In particular, as glycine metabolism has been shown to be important in skeletal muscle in obesity [26], the discrepancy between in vitro and in vivo results suggests that crosstalk between skeletal muscle and liver may play a key role in determining the availability and metabolic fate of glycine in the hepatocytes. "

Reviewer 3 Report

This manuscript does an excellent job of demonstrating the worsening effect of glycine in the liver of the obese mouse model. The authors clearly showed enhanced gluconeogenesis in the liver and reduced glucose homeostasis in glycine-supplemented obese mice. The overall paper is well written and the results are properly described. However, there are minor mistakes in the manuscript.

1. Ref. is missing for HFHS induced obese mice model.

2. Rewrite the sentence in p3, line#118-120.

Author Response

We appreciate your enthusiasm and positive assessment of the manuscript. Please find bellow our answers to your comments :

1. Ref. is missing for HFHS induced obese mice model : The reference « Beaulant et al. J Hepatol. 2022 » has been added.

2. Rewrite the sentence in p3, line#118-120 : The sentence and the paragraph have been corrected and completed.